# Systematic review and meta-analysis on the effect of depression on ART adherence among women living with HIV

Tadele Amare Zeleke[1]*, Kassahun Alemu[2], Tadesse Awoke Ayele[2], Zewditu Abdissa Denu[3], Lillian Mwanri[4], Telake Azale[5]

1 Department of Psychiatry, College of Medicine and Health Science, University of Gondar, Gondar, Ethiopia, 2 Department of Epidemiology and Biostatics, Institute of Public Health, College of Medicine and Health Sciences, University of Gondar, Gondar, Ethiopia, 3 Department of Anesthesia, College of Medicine and Health Sciences, University of Gondar, Gondar, Ethiopia, 4 Research Centre for Public Health, Equity and Human Flourishing, Torrens University Australia, Adelaide Campus, Adelaide, Australia, 5 Department of Health Promotion and Behavioral Sciences, Institute of Public Health, College of Medicine and Health Sciences, University of Gondar, Gondar, Ethiopia

* tadeleamare14@gmail.com

**Data Availability Statement:** All relevant data are within the paper and its supporting information files.

**Funding:** The author(s) received no specific funding for this work.

## Abstract

### Background

Depression is a very common psychiatric disorder in worldwide. Globally, Human Immuno-deficiency Virus (HIV) is highly prevalent among women, and are disproportionately affected by depression. Antiretroviral Therapy (ART) adherence which could highly be affected by depression is yet to be explored effectively. Depression affects overall poor HIV clinical outcomes, socioeconomic and social interactions. However, it is not well understood specifically how depression affects ART adherence in women living with HIV (WLWHIV). Investigating the effects of depression on ART adherence is critical in order to develop nuanced new evidence to address non-adherence in WLWHIV.

### Objective

To conduct a meta-analysis on the correlation between depression and adherence to antiretroviral therapy among women living with HIV in the globe.

### Method

Using population, exposed and outcome approach, we searched Scopus, PubMed, EMBASE, Cochrane Library, Psych info, Web of science and google scholar for cohort and cross-sectional studies globally. The search strategy was structured comprising terms associated with antiretroviral therapy and adherence, women living with HIV and depression. We evaluated the paper quality, using the Newcastle-Ottawa Scales (NOS). The fixed effect model was used to analysis the effect of depression on ART adherence.

### Result

A total of 8 articles comprise 6474 participants were included in this study. There were controversial findings related to the effect of depression to ART adherence. Among three cross-

**Competing interests:** All authors declare that no competing interest.

sectional study, one article demonstrating, depression was associated with ART adherence. Of the five cohort studies, four cohort studies reported association. The overall pooled estimated effect of depression on ART adherence was 1.02 [RR = 1.015 with 95% CI (1.004, 1.026)] with a p-value of 0.005.

## Conclusion and recommendation

Depression was the risk factor for ART adherence among women living with HIV. It is therefore, necessary for clinician to note this and perform screening for ART adherence.

## Trial registration

**The review protocol was developed with prospero registration:** CRD42023415935.

## Introduction

For over four-decades, HIV has continued to be a major public health concern globally affecting over 40.1 (33.6–48.6) million people, with two-thirds of PLWHIV reported to be in the African region [1–3]. The HIV infections and death rates have increased in recent times as a result of service disruptions and slowing of public health response to HIV due COVID-19 [1–3], and women comprise 51% of the global total number of PLWHIV [4, 5].

Human Immunodeficiency Virus and its complications, the Acquired Immunodeficiency Syndrome (AIDS) (sometimes referred as HIV/AIDS), impose a significant mental health burden on PLWHIV [6, 7]. The human immune deficiency virus induces immune activation in the brain that may lead to depletion and then reduce the level of serotonin, thereby risk of depressive symptoms and may change how the person thinks and behaves causing depression [6–8]. Additionally, stress associated with living with HIV can affect mental health, and some of the ARTs have also been implicated to have side effects that affect the person's mental health [9, 10]. Some predictors of mental health problems are associated with complex factors such as difficulties in accessing mental health services, loss of social support, poverty, loss of jobs, loss of relationships or death of loved ones, fear of disclosure of HIV status to others, poor access to medical treatment, and HIV stigma and discrimination [9, 10].

There is a "chicken-and/or-the-egg" dilemma in the relationship between depression and HIV disease progression, as these two have always had a bidirectional association masking which one caused the another. For example, while it has often been reported that HIV/AIDS causes depression [6, 7]. In a follow-up study depression predicted the progression of HIV/AIDS [11] and women who were diagnosed with psychiatric disorders were significantly associated with ART adherence than those who were not diagnosed with psychiatric disorders [12].

Generally it has been reported that chronic depression and stress might affect HIV disease progression [11], with the prevalence of depressive symptoms among PLWHIV reported to range from 12.8 to 78%, and the proportion of good adherence (>80%) ranging from 20 to 98% [13]. While the depressive symptoms rates in PLWHIV were reported to not vary by country's income group, the proportion of ART adherence varied with pooled rate of ART adherence in low-income and high-income countries reported to be 86% vs 67.5% respectively [13].

It has also been reported that WLWHIV are four times highly likely to suffer from depression compared to seronegative women due to experience of stigma and discrimination which leads to social isolation, loneliness and depression [14]. Women living with HIV suffer more

when compared to men [15, 16]. Additionally, depression can also be a mediating factor between HIV- related internal stigma and lower ART adherence, and can be amplified by poor social support/poor relationship in the family and increased loneliness [17, 18].

In WLWHIV, a highly likelihood of mental health problems and unmet needs correspond to directly reported link between depression and HIV stigma. For example, in a study by Armoon and colleagues, it was reported that participants who had been diagnosed with depression, were nearly twofold likely to report HIV stigma and discrimination, and that HIV stigma was associated with more complex mental health issues including anxiety and suicidal ideation [19]. Women may be particularly vulnerable to HIV stigma and its negative psychological effects [20] and social support had a moderating the relationship between stigma and depressive symptoms [21].

In depressed women, death was reported to be threefold likely than non-depressed women living with HIV [22]. Women living with HIV have also been reported to be at a higher risk of reduced uptake of ART, and at a higher incidence of treatment interruption than men [23–25]. However, equivocal reporting of WLWHIV having higher adherence to ART than men and reporting of no differences between men and women in ART adherence exists [26, 27]. Additionally, negative association between depression and ART adherence [12], and association between depression and ART non-adherence have also been reported [28–30] including in WLWHIV [31]. In the contrary, increased depressive symptoms scores were inversely associated with virologic failure, and immune system suppression [29], which are strongly associated with ART non-adherence [32]. However, in other studies, the contrary has been reported where depression was not statistically significantly associated with ART adherence and mortality [33].

Treatment of depression has been reported to increase ART adherence and CD4 T-cell responses [34]. The systematic review in developed countries showed that the women often depicted lower adherence to ART, and WLWHIV required specialized care to increase adherence to ART [23]. Early identification and treatment of mental health have been associated with improved ART adherence in other studies [35].

The above findings provide and equivocal outcomes of the effect of depression on ART adherence and better further studies need to be conducted to provide a clearer evidence of the effects of depression among WLWHIV. However, meta-analysis on the correlation between depression and ART adherence in WLWHIV were not done globally. Therefore, the current systematic review and meta-analysis investigates a meta-analysis on the correlation between depression and adherence to antiretroviral therapy among women living with HIV in the globe. The meta-analysis of the correlation between depression and adherence to antiretroviral therapy among women living with HIV is used to provide a new nuanced evidence to inform practice and policy frameworks.

## Methods

### Protocol registration and publication

Our systematic review and meta-analysis were registered on the International Prospective Register of Systematic Reviews (PROSPERO) with the number: CRD42023415935.

The protocol followed the Preferred Reporting Items for Systematic Reviews and Meta-Analysis(PRISMA) guidelines for methodological uniformity of the review process [36].

### Sources and data search strategy

The PEO approach was employed a systematic search strategy for this systematic review. The P (Population of interest) represented women living with HIV. The E (Exposure variable)

represented WLWHIV and depression. The O (Outcome) variable represented antiretroviral non-adherence. Electronic and manual searching were used to identify studies for the systematic review and meta-analysis. EDLINE, PubMed, psych INFO, EMBASE, CINAHL, Web of Science, Cochran library, and Scopus were searched to access the data. The key search terms were: (depression, OR "depression disorder" OR"depressive disorder" OR "major depression disorder" OR "major depressive disorder" OR "common mental disorder" OR "common mental illness" OR "mental illness" OR "mental health problem" OR "psychological distress" OR "psychological disorder" OR "chronic depression" OR "dysthymia disorder" AND "antiretroviral therapy" OR "highly active antiretroviral therapy" OR "ART compliance" OR "ART non-compliance" OR "antiretroviral therapy adherence" OR "optimal ART adherence" AND women OR woman OR femal* OR girl* OR lad* AND "HIV positive" OR "HIV infection" OR "Human Immune Virus" AND 2000/01/01 to 2022/12/30.

In systematic review, the studies reported in a cross-sectional study design and effect of depression on ART adherence in cohort study design were included. For meta-analysis, only cohort study findings were included because it showed cause and effect relationship between depression and ART adherence in WLWHIV. However, in cross-sectional study design either depression has effect on ART adherence or vice versa could not be known.

## Techniques for searching

Boolean operators like AND, and OR were used. Truncation: searches the base (trunk) of the word to include all variants of the word like femal* (female, females) and lad*(lady, ladies).

## Eligible criteria

**Inclusion criteria.** Peer reviewed cross-sectional and cohort study designed from across the world, published in English from 1 January 2000 to 30 December 2022 were included in the final analysis. Cross-sectional studies reporting an association between depression and ART adherence and cohort studies reporting the effect of depression on ART adherence were included.

## Exclusion

We excluded duplicated articles, reviews, commentaries, perinatal and postnatal women living with HIV.

## Selection process

In the first instance, all research articles obtained from the specified databases were exported to EndNote version 20. Duplication were excluded by using covidence. Titles and abstracts were screened, followed by full text screening. Further detailed information was obtained by emailing the authors for articles which did not have full information available publicly.

## Methods of data extraction and quality assessment

Two reviewers (SS and TK) evaluated the literature using the title and the abstract prior to reading the full-text of articles. Retrieved articles were further screened according to the pre-specified inclusion and exclusion criteria. We resolved the differences through a discussion between the two people screening the articles.

The standard format of the data extraction was used to identify studies, with the following information extracted: the first author, year of publication, study design, country where the

study was held, exposure (depression), outcome (ART adherence), sample size, AOR and RR, with 95% CI, and tool.

The quality of the included studies was evaluated using Newcastle-Ottawa Scales (NOS) for cohort and cross-sectional studies [37, 38]. The sample size representativeness, and comparability between participants, ascertainment of depression on ART adherence, and statistical quality were the domains of NOS used to assess each study's quality. Actual agreement and agreement beyond chance (unweighted Kappa) were evaluated by two reviewers' agreements. We considered the value 0 as poor agreement, 0.01 to 0.02 as slight agreement, 0.21 to 0.4 as fair agreement, 0.41 to 0.60 as moderate agreement, 0.61 to 0.80 as substantial agreement, and 0.81 to 1.00 as almost perfect agreement [39]. Therefore, in this review, the agreement beyond the chance was 0.75, which is a substantial agreement.

### Data synthesis and analysis

To explore the effect of depression on ART adherence in WLWHIV in the included studies, the logarithms of RR and the standard error of the logarithms of RR were computed. Data were exported to STATA version 14 analysis. The fixed effect model was used to show summary statistics, and the heterogeneity of the study was tested with $I^2$ and Q [40]. The thresholds for $I^2$ heterogeneity 25% as low, 50% as a medium, and 75% as high [41]. The assumptions that: there is a fixed effect model and that there is no random variation across studies were applied [40]. A small study bias was examined through symmetry and asymmetric funnel plot and objective inspection of Egger's regression test [42]. Publication bias was declared if the funnel plot was asymmetrical or if Egger's regression assumption test result was statistically significant (p<0.05) [43, 44]. The correlation between depression and ART adherence was presented at a 95% confidence level, and the result was described using narrative synthesis, tables and figures. Sensitivity analysis will be used to show a single study influences the overall findings. Certainty assessment: the authors are certain for the findings because of assessing the quality of papers with the independent ratter's, checking of heterogeneity tests, bias analysis and sensitivity analysis.

## Result

### Reporting of the studies

Preferred Reporting Items for Systematic Reviews and Meta-Analyses (PRISMA) guideline was used to report the selection process and reason for excluding papers in a systematic reviews and meta-analyses [36]. A total of 2643 articles were found in the systematic literature search of the available total, 403 articles were duplicates, 2240 were assessed as irrelevant after screening the title and abstract, four articles after reviewed full article (two articles stated the effect of depression on death, one article dealt with stress, and one article was due to small sample size(n = 13), and they were excluded from the analysis. In total, eight studies were included in the current systematic review. For the meta-analysis, only four articles, designed as cohort studies were included (Fig 1).

### Characteristics of the studies

Eight articles were included in this systematic review and meta-analysis. Three studies were conducted with a cross-sectional study design [45–47] and five studies were conducted with a cohort study design [17, 25, 48–50]. The risk of depression was analysed in RR for cohort and for cross-section studies AOR was reported as a significant factor. To show the correlation between depression and ART adherence, only cohort studies were included. However, cross-sectional studies were narrated in systematic review. Depression was assessed by the CES-D assessment too l (Table 1).

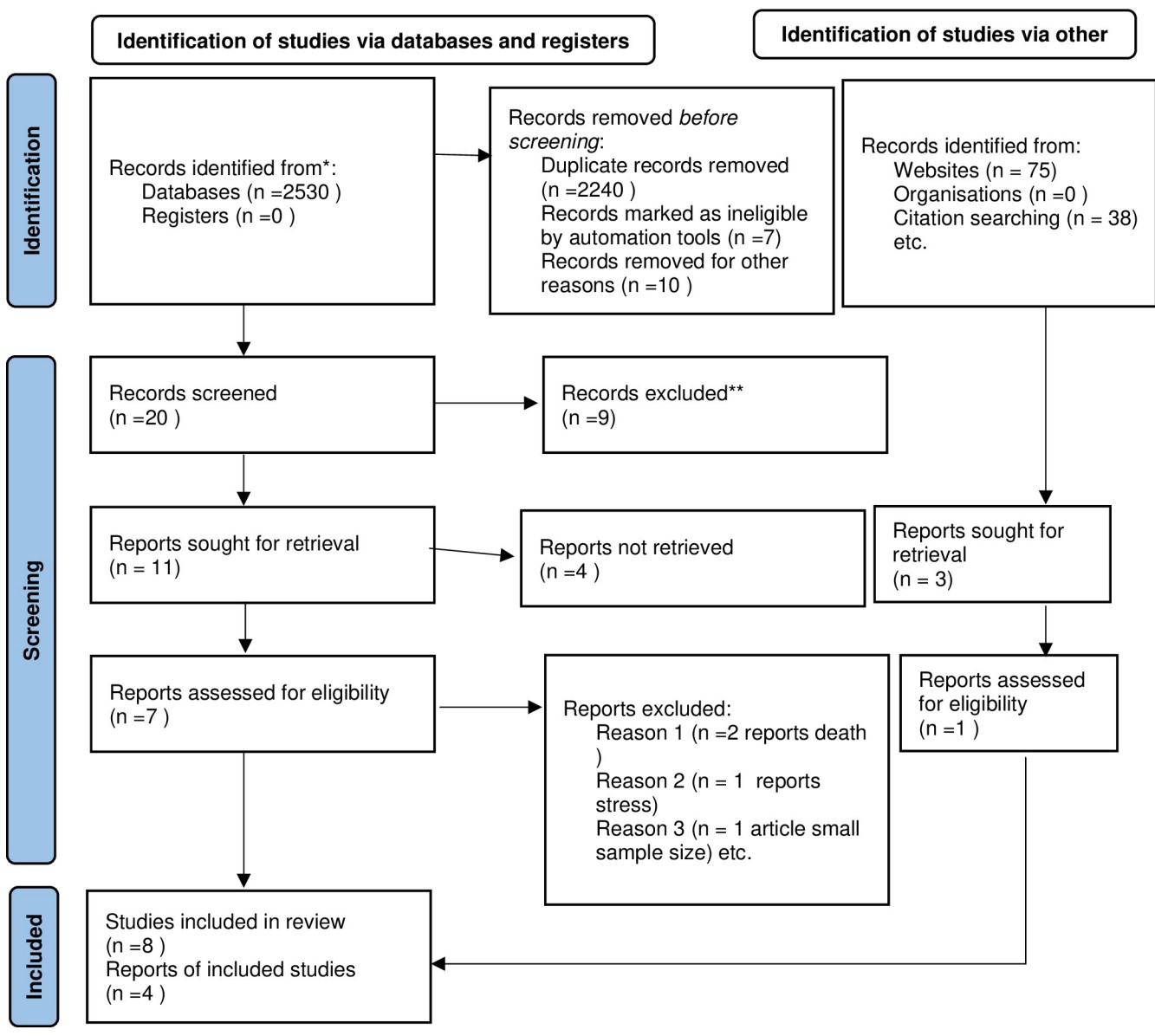

**Fig 1. PRISMA 2020 checklist showed how was the research articles were searched and selected, 2023.**

## Qualities of included studies

The Newcastle-Ottawa Scale (NOS) was used to assess the quality of the studies methodologically. In the evaluation, eight studies met the quality assessment in terms of selection, outcome measurement, and non-response bias. The Kappa value was calculated to assess the agreement between two raters. In this review, the agreement beyond the chance was 0.75 which is a substantial agreement [39] (Table 2).

## Sensitivity analysis

In the sensitivity analysis, there was no single study that was influencing the overall meta-analysis estimate (Fig 2).

**Table 1. Characteristics of the effect of depression on ART adherence among women living with HIV, 2023.**

| Publication Year | Author | Study design | Country | Exposure | Outcome | Sample | RR | AOR | Tool | Duration |
|---|---|---|---|---|---|---|---|---|---|---|
| 2011 [45] | Adeline Nyamathi | Cross-section | India | Depression | non-adherence | 68 | | No association | CES-D | |
| 2003 [25] | Barbara J | Cohort | New York State | Depression | non-adherence | 1827 | 1.92(1.00–3.68) | | CES-D | |
| 2019 [48] | Jon C. | Cohort | United State | Depression | non-adherence | 1491 | 1.54 (1.21 to 1.97 | | CES-D | 2013–2017 |
| 2016 [17] | Bulent Turan, | Cohort | US | Depression | non-adherence | 1019 | 1.04(1.02–1.05). | | CES_D | 2013–2014 |
| 2018 [46] | Adeline Nyamathi | Cross-section | India | Depression | non-adherence | 400 | | No association | CES-D | |
| 2016 [49] | Marcia McDonnell Holstad | Cohort | America | Depression | non-adherence | 193 | No association | | CES-D | 2005–2008 |
| 2017 [50] | Bulent Turan | Cohort | US | Depression | non-adherence | 1356 | 1.03[1.01–1.04] . | | CES-D-20 | 2014–2015 |
| 2019 [47] | Sandra Andinia, | Cross-sectional | Indonesia | Depression | non-adherence | 120 | | 3.64(1.7, 7.84) | CES-D | |

**Heterogeneity tests.** The $I^2$ and p-value were considered. The $I^2$ close to zero and the p-value greater than 0.05 showed that there was no heterogeneity [41]. Therefore, in this study, the ($I^2$ = 0.0% and the p-value was 0.778), which showed there was no heterogeneity among the studies. Hence fixed effect model was used for this meta-analysis (Fig 3).

**Galbraith plot.** Heterogeneity test with Galbraith plot (no heterogeneity because each study lies between CI (-2 to 2) (Fig 4).

## Publication bias

**Subjective assessment of publication bias.** No evidence of publication bias was found by the funnel plot test of the effect of depression on ART adherence among WLWHIV. The funnel plot appears symmetric suggested that no publication bias/small study effect (Fig 5).

**Objective assessment of publication bias.** Objective or statistical technique by using egger or Begg's test.

No evidence of publication bias was found by Egger's regression since the p-value was greater than 0.05, which was 0.139 (Table 3).

## The correlation between depression and adherence to antiretroviral therapy among women living with HIV

A total of eight studies comprise 6474 participants were included in this study. Of the three cross-sectional studies, two reported no association between depression and ART adherence, and one study reported an association between depression and ART adherence, with an AOR of 3.64(1.7, 7.84). In the five cohort studies that aimed to depict the effect of depression on

**Table 2. Showed that the kappa value of the agreement between the raters.**

| | | Value | Asymptotic Standardized Error[a] | Approximate T[b] | Approximate Significance |
|---|---|---|---|---|---|
| Measure of Agreement | Kappa | .750 | .226 | 2.191 | .028 |
| N of Valid Cases | | 8 | | | |

a. Not assuming the null hypothesis.

b. Using the asymptotic standard error assuming the null hypothesis.

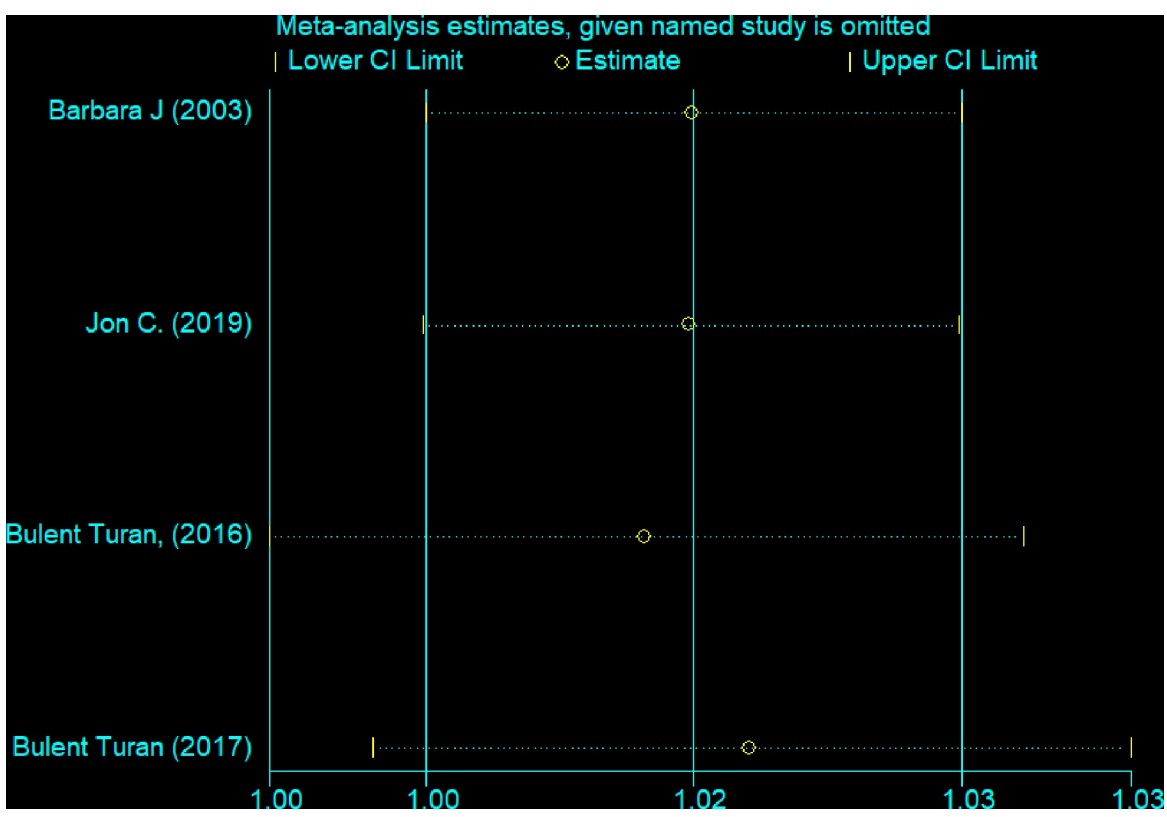

**Fig 2. There was no a single study influences other studies, 2023.**

ART adherence among WLWHIV, one study reported that depression had no effect on ART adherence, and four depicted that depression had an effect on ART adherence, with $I^2 = 0$ and p-value 0.778 (Fig 6). The inverse variance model was used, with the RR converted to LogRR and SELogRR form. The overall inverse variance the correlation between depression and adherence to antiretroviral therapy among women living with HIV was 1.02 [RR = 1.015 with 95% CI (1.004, 1.026)] with a p-value of 0.005, demonstrating that depression was the risk factor for ART adherence (Fig 6).

## Discussion

In this systematic review eight studies involving 6474 study participants were included. All study participants were WLWHIV. As we found no other studies conducted on this topic, we theorise that this study is the first systematic review and meta-analysis that has ever been conducted on an effect of depression on ART adherence among WLWHIV. From the eight studies, three were conducted with cross-sectional study design, and two of these studies depicted no association between depression and ART adherence [45, 46]. This finding was supported by other finding which showed that depression was not statistically significantly associated with ART adherence and mortality [33]. However, there has been one study that showed a strongly association of depression and ART adherence among WLWHIV with the AOR = 3.64 with 95% CI (1.7, 7.84 [47]).

Among the included five articles that were cohort study designed, one study stated that there was no association between depression and ART adherence [49]. Four studies revealed that depression was the risk factor for ART adherence among WLWHIV.

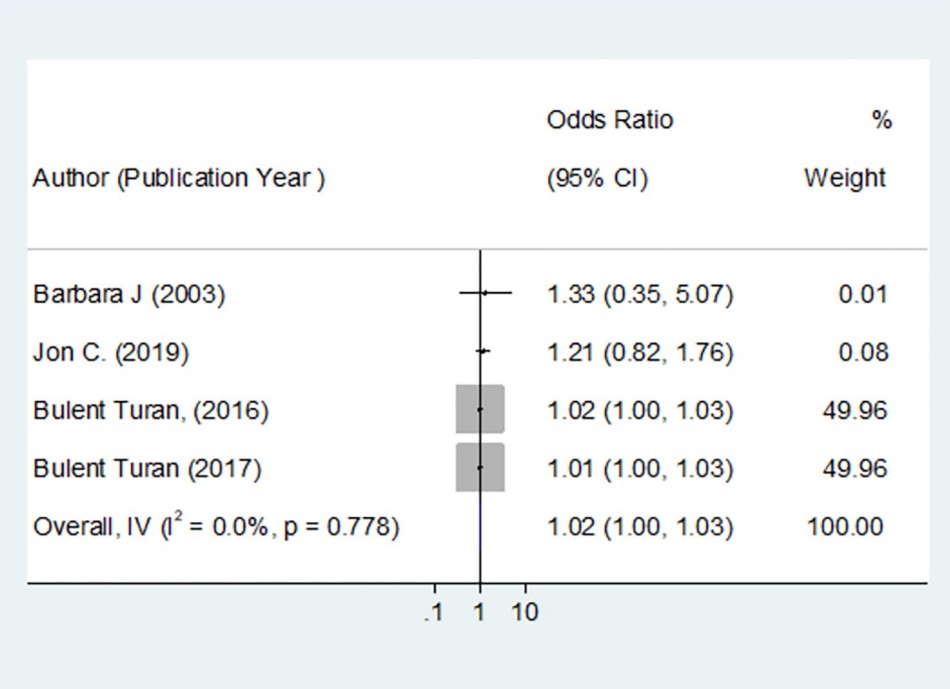

**Fig 3. The plot showed the heterogeneity test of effect of depression on ART adherence among women living with HIV, 2023.**

The overall correlation between depression and adherence to antiretroviral therapy among women living with HIV was 1.02 [RR = 1.015 with 95% CI(1.004, 1.026)] with a p-value of 0.005 [17, 25, 48, 50]. This finding was supported by other evidence [28–30], where depression has been reported to be associated with ART non-adherence. The possible reasons that have been postulated for these assertions is that women might be vulnerable to HIV- related stigma and its negative psychological problem, further leading to lost to follow-up [21]. It has also been acknowledged elsewhere that women are at twofold higher risk of reduced uptake of ART, and a higher incidence of treatment interruption than men due to HIV related stigma

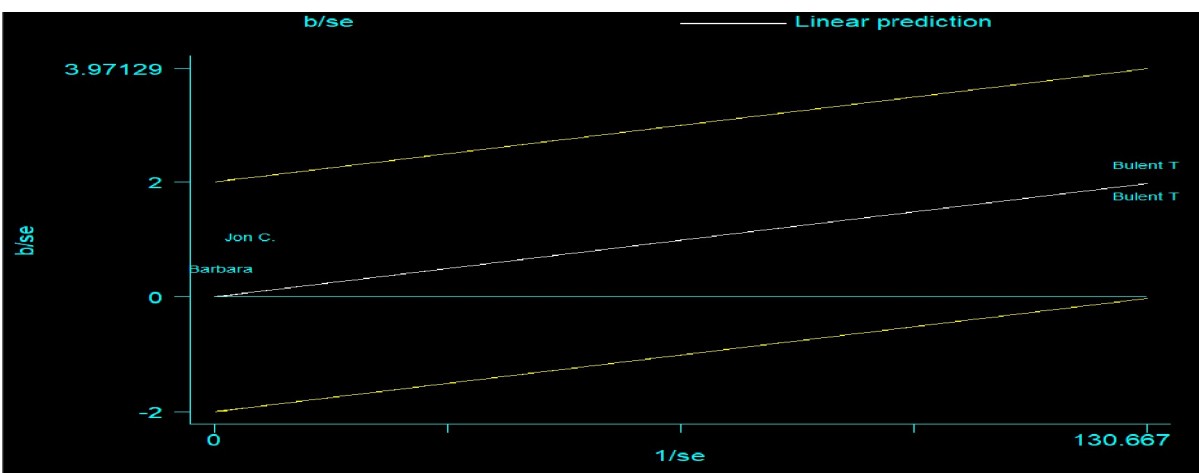

**Fig 4. Galbraith plot showed all studies laid between -2 and 2, hence no heterogeneity, 2023.**

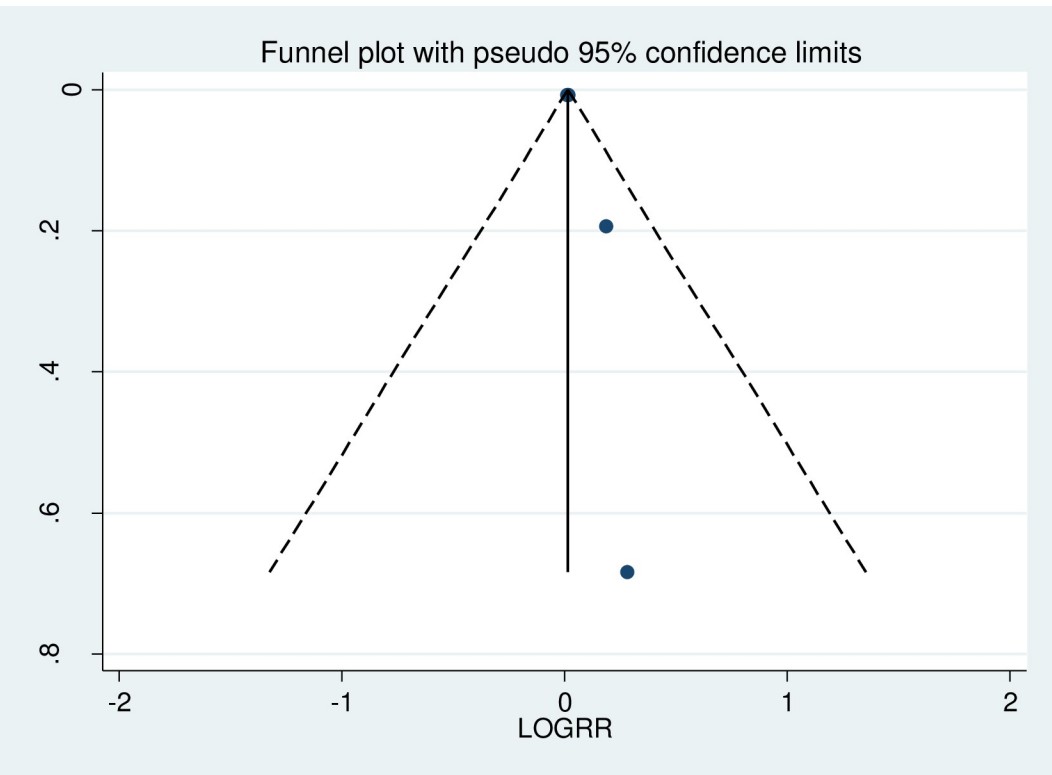

**Fig 5. The funnel plot showed the subjective assessment of publication bias, 2023.**

since HIV-related stigma in WLWHIV [19, 23]. Additional findings have reported a negative association between chronic depression and ART adherence over time [12, 31]. A systematic review conducted in developed countries has also shown that the women often depicted lower adherence to ART, compared with males, leaving WLWHIV to require specialized care services in order to increase adherence to ART [23] and improve the HIV and depression outcomes. Depression can also be among the reasons for women receiving poor social support/ poor relationship in the family, with the impact including increased loneliness and increased non- adherence to ARTs [17, 18].

About 64% of WLWHIV do not receive mental health services access which in turn, affects ART adherence [20]. Evidence has shown that depression is a predictor of HIV medication nonadherence in WLWHIV [31], and negative thoughts are more likely to occur about self and others in depressed subjects [51]. For example, depressed people have been reported to inaccurate, negative view of themselves, the world and the future, and it has been reported that due to negative thoughts, depressed individuals have more dysfunctional attitudes, more

**Table 3. Showed that the objectively assessment of publication bias, 2023.**

| Std.Eff | Coef. | Std. Err. | t | p>|t| | [95% Conf. Interval] |
|---------|-------|-----------|---|-------|----------------------|
| slope | 0.0099581 | 0.0029557 | 3.37 | 0.078 | -0.0027592–0.0226755 |
| bias | 0.6540891 | 0.2732074 | 2.39 | 0.139 | -0.5214274–1.829606 |

Number of studies = 4 Root MSE = 0.3766

Test of H0: no small-study effects p = 0.139

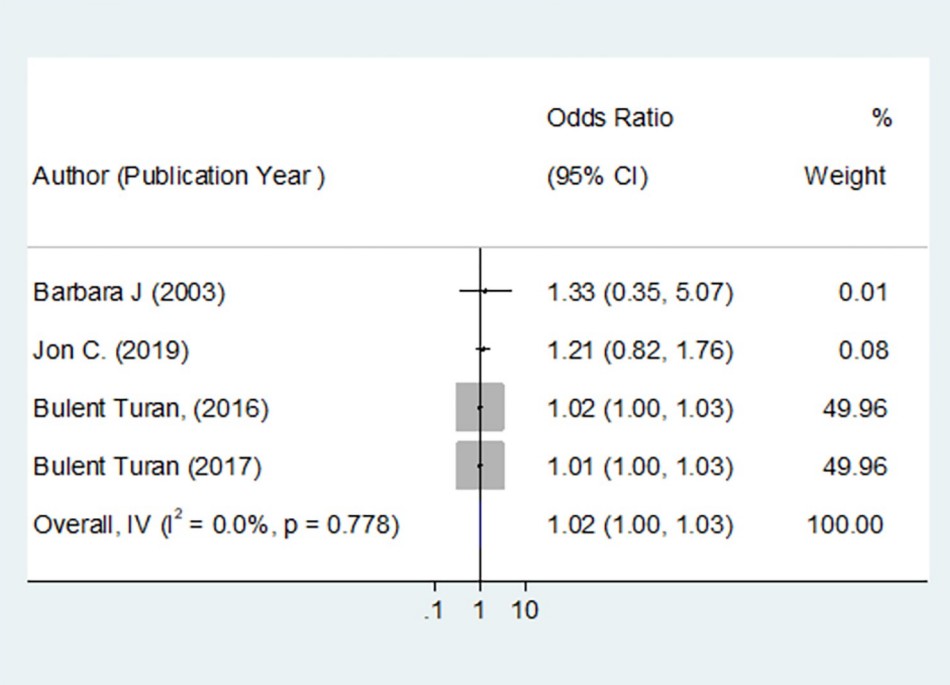

**Fig 6. The correlation between depression and adherence to antiretroviral therapy among women living with HIV, 2023.**

hopeless and more pessimistic ideas, all of which can be deterrent of up taking ART medication properly [52].

Conclusion and recommendation: in this systematic review and meta-analysis, there were controversial findings regarding the effect of depression on ART adherence. However, in the the correlation between depression and adherence to antiretroviral therapy among women living with HIV it was cleared shown that, depression was the risk factor for ART adherence in WLWHIV. Therefore, special attention pertaining these population is very important to improve the ART adherence. For additional information (S1 Checklist).

## Strengths and limitation of the review and meta-analysis

This systematic review and meta-analysis revealed the correlation between depression between ART adherence among WLWHIV and is considered untouched area of research.

**Limitation.** Articles published in other than the English language were not included, limitations that could have biased the overall outcomes. Three studies were conducted before one pill per day was not approved that enhanced poor ART adherence and only two studies were conducted in LMICs that could not be generalised for wider population.

## Implication of this finding

In the present study, the findings showed that there is the correlation between depression and ART adherence among women living with HIV. The most important advantage of the present study is integrating of mental health problems to develop a comprehensive study for the preliminary evaluation of depression in WLWHIV. This evidence that can support policymaker's decision to develop policy frameworks addressing depression including its negative impact on

ART adherence. Clinical service providers could also benefit from these findings through developing special practice drives that pay attention to depression in women living with HIV.

Future meta-analysis that will focus on both quantitative and qualitative studies address the correlation between depression and ART adherence are needed.

## Supporting information

**S1 Checklist. PRISMA 2020 checklist.**
(DOCX)

## Acknowledgments

The authors acknowledge the two reviewers for their critically review each article and the authors of the included studies.

## Author Contributions

**Conceptualization:** Tadele Amare Zeleke.

**Data curation:** Tadele Amare Zeleke, Zewditu Abdissa Denu, Lillian Mwanri.

**Formal analysis:** Tadele Amare Zeleke, Kassahun Alemu, Tadesse Awoke Ayele, Zewditu Abdissa Denu, Lillian Mwanri, Telake Azale.

**Methodology:** Tadele Amare Zeleke, Kassahun Alemu, Tadesse Awoke Ayele, Lillian Mwanri, Telake Azale.

**Software:** Kassahun Alemu, Tadesse Awoke Ayele, Telake Azale.

**Supervision:** Kassahun Alemu, Tadesse Awoke Ayele, Zewditu Abdissa Denu, Lillian Mwanri, Telake Azale.

**Validation:** Lillian Mwanri.

**Visualization:** Lillian Mwanri.

**Writing – review & editing:** Tadele Amare Zeleke, Kassahun Alemu, Tadesse Awoke Ayele, Zewditu Abdissa Denu, Lillian Mwanri, Telake Azale.

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
