## [Decision Letter · Decision Letter 0]

29 Jan 2024

PONE-D-23-42499Systematic review and meta-analysis on the effect of depression on ART adherence among women living with HIVPLOS ONE

Dear Dr. Zeleke,

Thank you for submitting your manuscript to PLOS ONE. After careful consideration, we feel that it has merit but does not fully meet PLOS ONE’s publication criteria as it currently stands. Therefore, we invite you to submit a revised version of the manuscript that addresses the points raised during the review process.

We look forward to receiving your revised manuscript.

Kind regards,

Amos Buh, BSc., MPH, PhD

Academic Editor

PLOS ONE

Additional Editor Comments:

Please address all comments from reviewers.

Reviewers' comments:

Reviewer's Responses to Questions

**Comments to the Author**

1. Is the manuscript technically sound, and do the data support the conclusions?

Reviewer #1: Yes

Reviewer #2: Yes

Reviewer #3: Partly

2. Has the statistical analysis been performed appropriately and rigorously? 

Reviewer #1: N/A

Reviewer #2: No

Reviewer #3: I Don't Know

3. Have the authors made all data underlying the findings in their manuscript fully available?

Reviewer #1: Yes

Reviewer #2: Yes

Reviewer #3: Yes

4. Is the manuscript presented in an intelligible fashion and written in standard English?

Reviewer #1: Yes

Reviewer #2: Yes

Reviewer #3: Yes

5. Review Comments to the Author

Reviewer #1: The manuscript can be accepted. This research is technically sound and well written. As such I recommend that it be published in this journal to enable the findings to be shared widely. Well dine to the authors

Reviewer #2: Reviewer’s Comment

Overview

The manuscript, entitled "Systematic Review and Meta-Analysis on the Impact of Depression on ART Adherence Among Women Living with HIV," is well-written. It addresses a crucial aspect of the HIV program, particularly in the context of improving adherence among women on ART. This work is significant due to its potential to influence strategies and interventions aimed at improving health outcomes for this vulnerable population. I have the following suggestions to improve the manuscript:

Abstract

• On page 2, line 34, the authors’ stated objective, “to pool the effect of depression on ART adherence among women living with HIV,” is generally clear but could be improved for precision and clarity. An improved objective might be: “To conduct a meta-analysis on the correlation between depression and adherence to antiretroviral therapy among women living with HIV in Sub-Saharan Africa.”

Introduction

• This section effectively introduces the topic and establishes the relevance of studying the effect of depression on ART adherence in women living with HIV. It might benefit from a more concise summary of existing literature and a clearer articulation of the knowledge gap this study addresses.

Objective

• On page 7, line 2, the objective is clearly stated and aligns well with the introduction. However, the gap the study was trying to address was not clearly described.

• Authors should ensure that the objective is clear and directly linked to the gaps or issues identified in the introduction. This creates a cohesive narrative from the problem to the study's aim.

• It was unclear why the authors stated the objective separately rather than integrating it with the introduction section. Authors should review journal guidelines and ensure this format aligns with journal requirements.

Method

• This section describes the systematic review process, including database search strategies and criteria for inclusion and exclusion of studies. It is detailed and provides a clear understanding of the methodology. However, in the introduction, (page 4, lines 86), the authors described the challenges of defining the etiology of the exposure variable. Therefore, it is important to be sure that all articles are describing the same exposure variable.

• The exposure variable “depression” was not well defined for the studies selected. A table summarizing the definition of depression for each of the eight articles studied will ensure that we are not comparing “apples with oranges.”

Data Analysis

The manuscript uses a fixed-effect model for the meta-analysis and employs the Newcastle-Ottawa Scale for the quality assessment of the included studies. While these are standard practices, there are several areas for potential improvement:

Choice of Meta-Analytic Model: The fixed-effect model assumes that the effect size is constant across all studies, which might not be the case here given the varied contexts of the studies. A random-effects model could provide a more generalizable estimate.

Heterogeneity Analysis: It's crucial to assess the heterogeneity among the included studies. If significant heterogeneity is present, exploring its sources (e.g., through subgroup analyses or meta-regression) can provide valuable insights.

Results

• The entire result section will need to be formatted to align with the style of other sections of the paper.

• The results are presented with sufficient detail, including the number of studies reviewed and the main findings. To improve this section, authors may provide a more nuanced analysis of the results, including subgroup analyses or meta-regressions if possible.

• Graphical representations like forest plots or funnel plots could be included to visually demonstrate the results and potential biases.

Conclusion and Recommendation

• The conclusions drawn align with the objective and results. Recommendations are practical and relevant. However, emphasizing the implications of these findings for future research and public health practice would enhance this section.

Reviewer #3: In this systematic review and meta-analysis, what you are trying to establish is the relationship of depression in WLWH and the ART adherence. The period of study is large (from 2000 to 2022) in which ART has changed a lot, going from high pill burden, less tolerable to more friendly treatments. In the abstract and introduction you state that there is not clear how depression affects ART adherence in WLWH. Although this is debatable, because there is an important amount of studies and information on this regard, my concern is that with this study and design this questions is not addressable; causality is very difficult to prove... so I recommend to rephrase and be more accurate with what the analysis is performed. In the introduction you also mention that mention that HIV affects the brian and the rest of the nervous system, causing depression, which is not accurate and the reference you are providing are old and not alined with that statement. Also in the Intro you use the term "women" "female" and "female gender"... please be consistent. Sex at birth is what normally is collected in clinical database, which nor necesarily is alined with the gender, very few cohorts address properly the difference.

Regarding the results and interpretation, I´m concerned that there are 3 studies before 2016 in which treatments were not 1 pill a day and harder to tlerate and also only 2 are from LMIC (India). These limitations are not addressed in the limitation section and therefore, very hard to extrapolate to wider population. Perhaps given these results you should give a perspective and propose more studies in LMIC on this regard.

The figure of the algorithm followed is not clear to me and for example, it is not shown that the NOS is applied.

6. PLOS authors have the option to publish the peer review history of their article (what does this mean?). If published, this will include your full peer review and any attached files.

Reviewer #1: No

Reviewer #2: No

Reviewer #3: No

---

## [Author Response · Author response to Decision Letter 0]

15 Feb 2024

Reviewer’s Comment

Overview

The manuscript, entitled "Systematic Review and Meta-Analysis on the Impact of Depression on ART Adherence Among Women Living with HIV," is well-written. It addresses a crucial aspect of the HIV program, particularly in the context of improving adherence among women on ART. This work is significant due to its potential to influence strategies and interventions aimed at improving health outcomes for this vulnerable population. I have the following suggestions to improve the manuscript:

Abstract 

• On page 2, line 34, the authors’ stated objective, “to pool the effect of depression on ART adherence among women living with HIV,” is generally clear but could be improved for precision and clarity. An improved objective might be: “To conduct a meta-analysis on the correlation between depression and adherence to antiretroviral therapy among women living with HIV in Sub-Saharan Africa.” 

Answer: Thank you so much for the comment. It is corrected as “To conduct a meta-analysis on the correlation between depression and adherence to antiretroviral therapy among women living with HIV in the globe. (page2, line 34-35)

Introduction

• This section effectively introduces the topic and establishes the relevance of studying the effect of depression on ART adherence in women living with HIV. It might benefit from a more concise summary of existing literature and a clearer articulation of the knowledge gap this study addresses. 

• Answer: Thank you so much for the comment: It is synthesized (page5 and 6, line 99-138)

Objective

• On page 7, line 2, the objective is clearly stated and aligns well with the introduction. However, the gap the study was trying to address was not clearly described. 

• Authors should ensure that the objective is clear and directly linked to the gaps or issues identified in the introduction. This creates a cohesive narrative from the problem to the study's aim.

• It was unclear why the authors stated the objective separately rather than integrating it with the introduction section. Authors should review journal guidelines and ensure this format aligns with journal requirements. 

• Answer: corrected and merged to introduction (page6, line 132-138)

Method

• This section describes the systematic review process, including database search strategies and criteria for inclusion and exclusion of studies. It is detailed and provides a clear understanding of the methodology. However, in the introduction, (page 4, lines 86), the authors described the challenges of defining the etiology of the exposure variable. Therefore, it is important to be sure that all articles are describing the same exposure variable. 

Answer: Thank you so much for the comments. for systematic review both a cross-sectional and cohort study designs were used. However, for meta-analysis only cohort studies were included. Because cohort studies showed the cause and effect relationship. page 7-8 and line 162-166

• The exposure variable “depression” was not well defined for the studies selected. A table summarizing the definition of depression for each of the eight articles studied will ensure that we are not comparing “apples with oranges.”

Answer: Thank you so much for the comments and suggestions. Both cross-sectional and cohort study designs were included in the systematic review only. However, to meta-analysis only cohort studies were included that showed the correlation between depression and ART adherence among women living with HIV which is stated in (page 7-8 and line 162-166) and (page 14 and line 284-290)

Data Analysis

The manuscript uses a fixed-effect model for the meta-analysis and employs the Newcastle-Ottawa Scale for the quality assessment of the included studies. While these are standard practices, there are several areas for potential improvement:

Choice of Meta-Analytic Model: The fixed-effect model assumes that the effect size is constant across all studies, which might not be the case here given the varied contexts of the studies. A random-effects model could provide a more generalizable estimate.

Answer: Thank you so much the detail comments. In meta-analysis, random effect model is implemented when there is high heterogeneity (I2>75% and p-value<0.05) which is stated in page 9 and line 207-208. In this meta-analysis, heterogeneity test was tested with I2 and P-value which was (I2= 0.0% and p-value 0.778) which showed low heterogeneity. Therefore, fixed effect model can be implemented. 

Heterogeneity Analysis: It's crucial to assess the heterogeneity among the included studies. If significant heterogeneity is present, exploring its sources (e.g., through subgroup analyses or meta-regression) can provide valuable insights.

Answer: Thank you so much for the comments. In this meta-analysis there was low heterogeneity (I2=0.0% and p-value=0.778) (page13 and line 258-263). Galbraith plot was used to test heterogeneity, from the plot no heterogeneity because each study lies between CI (-2 to 2) (page13, and line265-266). The plot can be found in the separated figure section (Fig3 and Fig4). 

Results

• The entire result section will need to be formatted to align with the style of other sections of the paper.

Answer: Thank you so much for the concerns. The authors prepared the manuscript according to the guideline. 

• The results are presented with sufficient detail, including the number of studies reviewed and the main findings. To improve this section, authors may provide a more nuanced analysis of the results, including subgroup analyses or meta-regressions if possible. 

Answer: Thank you so much for the comments. Since there was no heterogeneity: subgroup analysis and meta-regression were not done

• Graphical representations like forest plots or funnel plots could be included to visually demonstrate the results and potential biases.

Answer: Than you so much the comments. The figures were submitted in the separated section (Fig5) (page14 and line 274). The objective assessment of publication bias was tested that showed no publication bias(p-value=0.139) page14 and line 275-280(Table3) 

Conclusion and Recommendation

• The conclusions drawn align with the objective and results. Recommendations are practical and relevant. However, emphasizing the implications of these findings for future research and public health practice would enhance this section. 

Answer: Thank you so much for the comments. It is corrected in page18 and line350-360

Reviewer #1: The manuscript can be accepted. This research is technically sound and well written. As such I recommend that it be published in this journal to enable the findings to be shared widely. Well dine to the authors

Answer: Thank you so much

Reviewer #3: In this systematic review and meta-analysis, what you are trying to establish is the relationship of depression in WLWH and the ART adherence. The period of study is large (from 2000 to 2022) in which ART has changed a lot, going from high pill burden, less tolerable to more friendly treatments. 

Answer: Thank you so much for the comments. We added in the limitation section as” Three studies were conducted before one pill per day was not approved that enhanced poor ART adherence and only two studies were conducted in LMICs that could not be generalised for wider population”(Page 18 and line 347-349)

 In the abstract and introduction you state that there is not clear how depression affects ART adherence in WLWH. Although this is debatable, because there is an important amount of studies and information on this regard, my concern is that with this study and design this questions is not addressable; causality is very difficult to prove... so I recommend to rephrase and be more accurate with what the analysis is performed.

Answer: Thank you so much for the comments. In method section the following statement is added” In systematic review, the studies reported in a cross-sectional study design and effect of depression on ART adherence in cohort study design were included. For meta-analysis, only cohort study findings were included because it showed cause and effect relationship between depression and ART adherence in WLWHIV. However, in cross-sectional study design either depression has effect on ART adherence or vice versa could not be known” (page7 and 8 line162-166) and in result section (page11, and line 229-230), page14 line287-290) that showed only cohort study were included because cause and effect relationship is addressed with this study design. 

 In the introduction you also mention that mention that HIV affects the brian and the rest of the nervous system, causing depression, which is not accurate and the reference you are providing are old and not alined with that statement. 

Answer: Thank you so much for your comments. The statement is modified as “The human immune deficiency virus induces immune activation in the brain that may lead to depletion and then reduce the level of serotonin, thereby risk of depressive symptoms and may change how the person thinks and behaves causing depression” (page4 and line78-81).

Also in the Intro you use the term "women" "female" and "female gender"... please be consistent. Sex at birth is what normally is collected in clinical database, which nor necesarily is alined with the gender, very few cohorts address properly the difference.

Answer: Thank you so much for your comments. It is corrected throughout the document. 

Regarding the results and interpretation, I´m concerned that there are 3 studies before 2016 in which treatments were not 1 pill a day and harder to tlerate and also only 2 are from LMIC (India). These limitations are not addressed in the limitation section and therefore, very hard to extrapolate to wider population. Perhaps given these results you should give a perspective and propose more studies in LMIC on this regard.

Answer: Thank you so much for the comments. We accepted and added in the limitation section as. “Three studies were conducted before one pill per day was not approved that enhanced poor ART adherence and only two studies were conducted in LMICs that could not be generalised for wider population” (page18 and line 347-349)

The figure of the algorithm followed is not clear to me and for example, it is not shown that the NOS is applied. Answer: Thank you so much for the comment. The authors stated about NOS in the method section as “The sample size representativeness, and comparability between participants, ascertainment of depression on ART adherence, and statistical quality were the domains of NOS used to assess each study’s quality. Actual agreement and agreement beyond chance (unweighted Kappa) were evaluated by two reviewers’ agreements” (page9 and line 194-198) and in the result section specially in “Qualities of included studies” (page12-13 and line 246-253, and Table2)

---

## [Decision Letter · Decision Letter 1]

23 Feb 2024

Systematic review and meta-analysis on the effect of depression on ART adherence among women living with HIV

PONE-D-23-42499R1

Dear Dr. Zeleke,

We’re pleased to inform you that your manuscript has been judged scientifically suitable for publication and will be formally accepted for publication once it meets all outstanding technical requirements.

Kind regards,

Amos Buh, BSc., MPH, PhD

Academic Editor

PLOS ONE

Additional Editor Comments (optional):

Reviewers' comments:

Reviewer's Responses to Questions

**Comments to the Author**

1. If the authors have adequately addressed your comments raised in a previous round of review and you feel that this manuscript is now acceptable for publication, you may indicate that here to bypass the “Comments to the Author” section, enter your conflict of interest statement in the “Confidential to Editor” section, and submit your "Accept" recommendation.

Reviewer #1: All comments have been addressed

2. Is the manuscript technically sound, and do the data support the conclusions?

Reviewer #1: Yes

3. Has the statistical analysis been performed appropriately and rigorously? 

Reviewer #1: N/A

4. Have the authors made all data underlying the findings in their manuscript fully available?

Reviewer #1: Yes

5. Is the manuscript presented in an intelligible fashion and written in standard English?

Reviewer #1: Yes

6. Review Comments to the Author

Reviewer #1: The authors have adequately addressed all the review recommendations. The manuscript is ready for publishing.

7. PLOS authors have the option to publish the peer review history of their article (what does this mean?). If published, this will include your full peer review and any attached files.

Reviewer #1: No

---

## [Editor Report · Acceptance letter]

27 Mar 2024

PONE-D-23-42499R1 

PLOS ONE

Dear Dr. Zeleke, 

I'm pleased to inform you that your manuscript has been deemed suitable for publication in PLOS ONE. Congratulations! Your manuscript is now being handed over to our production team.

Kind regards, 

on behalf of

Dr. Amos Buh 

Academic Editor

PLOS ONE